# Age-Related Characteristics of Diastolic Dysfunction in Type 2 Diabetes Patients

**DOI:** 10.3390/jcm14165772

**Published:** 2025-08-15

**Authors:** Elena-Daniela Grigorescu, Bogdan-Mircea Mihai, Georgiana-Diana Cazac-Panaite, Adina-Bianca Foșălău, Alina Onofriescu, Mariana Floria, Cristina Gena Dascălu, Alexandr Ceasovschih, Laurențiu Șorodoc, Cristina-Mihaela Lăcătușu

**Affiliations:** 1Unit of Diabetes, Nutrition, and Metabolic Diseases, Faculty of Medicine, “Grigore T. Popa” University of Medicine and Pharmacy, 700115 Iasi, Romania; elena-daniela-gh-grigorescu@umfiasi.ro (E.-D.G.); georgiana-diana_cazac@umfiasi.ro (G.-D.C.-P.); giba.adina-bianca@d.umfiasi.ro (A.-B.F.); alina.onofriescu@umfiasi.ro (A.O.); cristina.lacatusu@umfiasi.ro (C.-M.L.); 2Clinical Centre of Diabetes, Nutrition and Metabolic Diseases, “Sf. Spiridon” County Clinical Emergency Hospital, 700111 Iasi, Romania; 3Internal Medicine, “Grigore T. Popa” University of Medicine and Pharmacy, 700115 Iasi, Romania; floria.mariana@umfiasi.ro (M.F.); alexandr.ceasovschih@umfiasi.ro (A.C.); laurentiu.sorodoc@umfiasi.ro (L.Ș.); 4Department of Internal Medicine, “St. Spiridon” County Clinical Emergency Hospital, 700111 Iasi, Romania; 5Discipline of Medical Informatics and Biostatistics, Faculty of Medicine, “Grigore T. Popa” University of Medicine and Pharmacy, 700115 Iasi, Romania; cristina.dascalu@umfiasi.ro

**Keywords:** elderly, type 2 diabetes mellitus (T2DM), left ventricular diastolic dysfunction (LVDD), inflammatory markers

## Abstract

**Background**: Asymptomatic left ventricular diastolic dysfunction (LVDD) occurs in type 2 diabetes mellitus (T2DM) patients, particularly among the elderly. **Aim**: This study aimed to identify diastolic function changes over a 52-week follow-up and the predictive factors for LVDD in T2DM patients without atherosclerotic manifestations. **Methods**: Diastolic function, metabolic profile, atherogenic indexes, and subclinical inflammatory markers were assessed at baseline and after one year in 138 T2DM outpatients. All variables were compared in patients with and without LVDD across three age groups. **Results**: The patients were 57.86 ± 8.82 years old, 49.3% male, with a mean 5-year diabetes duration and a median HbA_1c_ of 7.8%. At baseline, 71 patients had grade 1 LVDD, 12 had grade 2 and 3 LVDD, and 15 had indeterminate LVDD. In the elderly group, 29 patients had LVDD. The logistic regression analysis identified age over 65 as an independent risk factor for LVDD (Exp B = 9.85, 95% CI: 1.29–75.36, *p* = 0.027). LVDD patients had a longer diabetes duration and a higher prevalence of diabetic neuropathy. Elderly patients had the lowest E/A, e’, lateral s’, atherogenic and Castelli risk indexes, and significantly higher E/e’, EDT, LAVI and TNF-alpha values (*p* < 0.05). After 52 weeks, diastolic function worsened in 27 patients, who had no significant differences compared to those with stable or improved diastolic function. **Conclusions**: LVDD was common in our T2DM patients without known cardiovascular disease, and age increases the LVDD risk. Echocardiographic assessment is necessary, especially in elderly T2DM patients with co-morbidities, to identify patients at risk of progression to heart failure early.

## 1. Introduction

Cardiovascular disease (CVD) is the leading cause of death in patients with diabetes. Early detection of subclinical cardiac dysfunction, allowing the initiation of therapies with proven cardiac benefits, can prevent the onset and progression of overt CVD, including heart failure with preserved ejection fraction (HFpEF), and therefore the risk of major cardiovascular events [1,2].

Left ventricular diastolic dysfunction (LVDD) has been recognized as an early manifestation of diabetic cardiomyopathy. Incipient structural and functional cardiac impairment develops without clinical symptoms. These changes are promoted by metabolic disturbances (decreased glucose oxidation, increased free fatty acids), impaired cellular function (inadequate calcium handling, augmented oxidative stress, mitochondrial dysfunction), structural alterations (advanced glycation end-products, cardiomyocyte hypertrophy), and endothelial dysfunction associated with systemic inflammation. Additionally, the activation of the renin–angiotensin–aldosterone system and autonomic neuropathy lead to fibrosis and cardiomyocyte stiffness [3,4,5]. Such impairments can occur independently of other causes or conditions (like hypertension, coronary artery disease, and obesity) and reflect the impact of diabetes on the myocardial structure and function [6].

Echocardiography is the most commonly used method to examine cardiac function in patients with diabetes. LVDD assessment should follow the latest 2016 European Association of Cardiovascular Imaging and American Society of Echocardiography (EACVI/ASE) recommendations, which consider conventional echocardiography and tissue Doppler imaging as essential techniques for effectively identifying diastolic dysfunction. Echocardiographic parameters such as left atrium volume (LAV) index, tricuspid regurgitation velocity, early (E) mitral inflow velocity to early diastolic velocity (e’) ratio, and septal or lateral e′ velocity are included in an algorithm that diagnoses and grades LVDD in situations where the left ventricular ejection fraction (LVEF) is preserved. LVDD grade can be established if three or four criteria are met [7,8].

The prevalence of LVDD among patients with type 2 diabetes mellitus (T2DM) without overt CVD ranges from 25% to 70%. Longitudinal studies including such patients have identified predictors of LVDD deterioration, such as advanced age, retinopathy, hypertension, obesity, albuminuria, poor glycemic control, and glycemic variability [9]. Other research attempts have focused on biomarkers such as asprosin, cystatin C, galectin-3, plasma endothelin-1, or N-terminal pro-B-type natriuretic peptide (NT-proBNP), which are typically associated with diastolic dysfunction in T2DM, aiming to find those high-risk patients with subclinical cardiac impairment [10,11,12,13]. Identifying cardiovascular and non-cardiac risk factors that contribute to the transition from asymptomatic to overt heart failure (HF) can facilitate the early implementation of comorbidity-targeted therapeutic strategies to delay HF progression.

Inflammatory markers, insulin resistance, and atherogenic indexes are potentially related to LVDD development and progression to overt HF [14]. The simple existence of diabetes and obesity can be associated with low-grade chronic inflammation. Neutrophil-to-lymphocyte ratio (NLR), easily determined from a complete blood count, appears to be another reliable marker of systemic inflammation, which several researchers have already analyzed in relation to diastolic parameters [15].

Understanding the varying data that have been communicated about factors predicting diastolic dysfunction in T2DM patients requires careful consideration to ensure an accurate interpretation. Challenges often occurring during a comprehensive investigation, such as the clinician’s experience, the equipment used for cardiac function assessment, diagnostic algorithm choices, inter-observer variability, study design, patient demographics, and associated comorbidities, can impact the results.

In routine clinical practice, many clinicians may either underestimate the importance of early detection of diastolic dysfunction among diabetes-related complications or lack the necessary methods to identify it.

Asymptomatic LVDD is a component of preclinical cardiac impairment and leads to worse outcomes of CVD. Besides age, obesity, hypertension, coronary artery disease, atrial fibrillation, and peripheral vascular disease, diabetes counts among the factors that can determine the progression from asymptomatic to symptomatic HF stages [16]. T2DM patients are considered to belong to the first stage of HF [17]. Implementation of the recent International Diabetes Federation (IDF) Global Clinical Practice Recommendations for Managing Type 2 Diabetes into practice requires performing early and regular screening using risk scores, symptom assessment, electrocardiography, echocardiography, and biomarker assessment [18].

An E/e’ ratio above 9 and right ventricle (RV) systolic pressure > 35 mm Hg, both independently predictive factors for HFpEF, are included in the H_2_FPEF Score. While LVDD is a central element of HFpEF, it is important to acknowledge that the left ventricle (LV) relaxation and compliance decline with normal ageing [19], secondary to unfavourable structural cardiac remodelling. Nagueh et al. consider that the echocardiographic variables E/A, ETD, and IVRT, have age-dependent limitations, while E/e’ ratio is less age-dependent (values over 14 are rarely identified in healthy persons) [7,8]. Healthy older people might also have undetected coronary diseases or other subclinical disorders, leading to a wider range of normal values. Similar filling patterns were reported in healthy elderly and in patients aged 40 to 60 years identified with mild diastolic dysfunction. The diastolic dysfunction of healthy, but elderly individuals could be explained by the increased stiffness of the left ventricle compared to younger people. A slower myocardial relaxation may also occur in older individuals, thus explaining the lower values of the E/A ratio, but data on ageing and relaxation are not entirely consistent across studies [7]. Moreover, some parameters cannot be determined or should not be measured in patients with atrial fibrillation or flutter, pericardial constriction, heavy mitral annular calcification, prosthetic mitral valve, surgical ring, and mitral valve disease [8].

In the community-based Framingham Heart Study, age was found to have positive associations with E/e’ ratio and negative associations with E/A ratio and e’ velocity [20]. However, diastolic dysfunction was defined in this study using the Olmstead criteria, which were applicable before the 2016 recommendations. Hence, the researchers suggested that using age- and sex-specific reference limits in the evaluation of diastolic function in clinical practice would identify a closer relationship between the diastolic dysfunction and other modifiable risk factors than age [8,16]. Other community-based studies hypothesized that mild dysfunction identified in persons aged 85 years or more could be interpreted as a phenotype of the ageing heart [21].

While changes in the cardiac structure, physical inactivity, deceleration of electrical conduction, and subclinical sinus node dysfunction are all contributors to the ageing-related diastolic dysfunction, the progression of diastolic dysfunction can also be accelerated by the coexistence of hypertension, obesity, diabetes, atherosclerosis, coronary disease, and chronic kidney disease [22,23]. These comorbidities are also age-related, augment the cardiac dysfunction mechanisms and elevate susceptibility to HF. Fang et al. previously detailed the complexity of the cardiac, vascular, metabolic and inflammaging mechanisms that exacerbate diastolic dysfunction [23].

Growing interest exists in the processes and pathways that contribute to age-related cardiac deterioration and lead to heart failure. All processes mentioned above (fibrosis, inflammation, mechanical stiffening, diastolic dysfunction, mitochondrial dysfunction, cardiomyocyte apoptosis and loss of regenerative capacity) seem to be driven by molecular mechanisms that include telomere shortening, somatic mutations, epigenetic changes, alterations in noncoding RNAs regulation of gene expression and senescence-associated secreted factors consisting in pro-inflammatory cytokines [24,25].

A monomorph, age-related-only phenotype of cardiac dysfunction is difficult to draft, given the multiple molecular pathways that are simultaneously involved in the complex array of heart ageing. Patients with diabetes exacerbate many of these molecular circuits and alter once more the metabolic environment of their cardiomyocytes due to the deleterious effects of hyperglycemia and insulin resistance [26,27].

Better knowledge of factors predicting LVDD presence and progression could facilitate the implementation of a screening tool allowing its early diagnosis among people with diabetes and no overt CVD. These steps could help clinicians promote therapeutic and lifestyle interventions (exercise, dietary plans) to counteract the progression of diastolic dysfunction [23].

To the best of our knowledge, the prevalence of HF in T2DM patients has been variously reported in different populations of our country, ranging from 30.7% in the FIND study to 42.69% in the elderly population [28,29]. The prevalence of grade 1 LVDD ranged between 54% and 73.5% in a few cross-sectional studies focusing on T2DM patients with MASLD and overt CVD [30,31].

The aims of this study were: (a) to establish the prevalence of LVDD in T2DM patients without atherosclerotic manifestations, (b) to investigate the hypothesis that metabolic parameters and inflammatory markers can predict diastolic dysfunction changes after 52 weeks, and (c) to identify the presence of these factors in patients with or without LVDD that belong to various age groups, including elderly.

## 2. Materials and Methods

### 2.1. Participants’ Enrolment

This is a post hoc analysis of a set of cross-sectional data collected at baseline and after 52 weeks from a series of consecutive outpatients with T2DM, as part of a prospective study conducted between June 2016 and February 2018 at the Clinical Centre of Diabetes, Nutrition, and Metabolic Diseases in Iași. All assessments were performed in accordance with the ethical standards of the Helsinki Declaration and approved by the Ethics Committees of “Grigore T. Popa” University of Medicine and Pharmacy Iași (3 April 2016) and of the “St. Spiridon” Emergency Clinical Hospital Iași (63274/16 December 2015). All patients signed an informed consent form before any procedure was performed.

The main inclusion criterion was suboptimal glycemic control (HbA_1c_ > 7%) in 35- to 80-year-old T2DM patients who were previously treated with metformin and/or sulfonylurea or acarbose, and for whom treatment was supplemented with sitagliptin, saxagliptin or exenatide at study admission. Patients with any acute or chronic form of atherosclerotic CVD (e.g., myocardial infarction, angina, coronary revascularisation, electrocardiographic findings of ischemia, stroke, transient ischemic attack, and peripheral arterial disease) were ineligible for enrollment. Other exclusion criteria were active smoking, insulin therapy, type 1 or secondary pancreatic diabetes, valvular heart disease, dysrhythmias, cardiac pacemakers, other inflammatory and/or severe conditions (pancreatitis, liver failure/viral hepatitis, gastrointestinal and kidney disease, malignancies), psychiatric disorders, serum triglycerides above 400 mg/dL, and uncontrolled blood pressure despite antihypertensive therapy (values above 140/90 mmHg).

All patients underwent the following procedures at baseline and after 52 weeks: collection of medical history data, standard clinical examination, screening for diabetes-related chronic complications, echocardiography, blood tests, and urine analysis.

To identify the characteristics of elderly individuals, all studied variables were compared between three age groups (<50 years, 50–64 years, and ≥65 years) at both the baseline and 52-week visits.

### 2.2. Clinical and Biochemical Parameters

Data on age, diabetes duration, chronic medication, and comorbidities were noted. The physical examination included anthropometric parameters (height, weight, waist circumference—WC, and calculation of body mass index—BMI), heart rate and blood pressure measurements, neuropathy assessment tests, and fundoscopy for all enrolled subjects. A resting electrocardiogram (ECG) was performed for all subjects.

After a minimum eight-hour fast, blood samples were drawn to evaluate the fasting plasma glucose (FPG), lipid profile (total cholesterol, high-density lipoprotein cholesterol—HDL-cholesterol, low-density lipoprotein cholesterol—LDL-cholesterol, triglycerides—TG), uric acid, liver and kidney function tests. HbA_1c_ was measured using ion-exchange high-performance liquid chromatography (Bio-Rad D-10™, Bio-Rad Laboratories Diagnostics Group, Hercules, CA, USA). A complete blood cell count, including WBCs, neutrophils, lymphocytes, and platelets, was also obtained. The glomerular filtration rate (GFR) was calculated using the Chronic Kidney Disease Epidemiology Collaboration (CKD-EPI) formula. Non-HDL cholesterol was assessed as the difference between total cholesterol and HDL-cholesterol. Neutrophil to lymphocyte ratio (NLR) and platelet to lymphocyte ratio (PLR) were also calculated. Another blood sample was centrifuged at 3000× *g* for 5 min, and the serum was stored at −20 °C for up to three months for immunological assays. Highly sensitive C-reactive protein (hsCRP), interleukin-6 (IL-6), tumour necrosis factor alpha (TNF-α), insulin, and C-peptide levels were measured using chemiluminescence techniques (IMMULITE 1000 Immunoassay System, Siemens Healthineers, Erlangen, Germany). The urine albumin-to-creatinine ratio (UACR) was determined from a first morning urine sample.

Insulin resistance indixes were calculated using the validated formulae [32]:Homeostatic Model Assessment for Insulin Resistance (HOMA-IR) = FPG (mg/dL) × insulinemia (μU/mL)/405;HOMA C-peptide = FPG (mg/dL)/18 × C-peptide (ng/mL) × 3.003)/22.5;Index C-peptide = 20/[C-peptide (ng/mL) × 3003 × FPG (mg/dL)/18)];Metabolic Score for Insulin Resistance (METS-IR) = ln [2 × FPG (mg/dL) + TG (mg/dL)] × BMI/[ln(HDL-cholesterol (mg/dL)] [33].Triglyceride-Glucose Index (TyGi) = ln[TG (mg/dL) × FPG (mg/dL)]/2 [34].

A series of composite lipid indices were also calculated using the following formulas applied to values converted to mmol/L [35]:Atherogenic Index of Plasma (AIP) = ln(TG/HDL-cholesterol)Atherogenic Index (AI) = non-HDL-cholesterol/HDL-cholesterolLipoprotein Combine Index (LCI) = (total cholesterol × TG × LDL-cholesterol)/HDL-cholesterolCastelli risk index-I (CRI-I) = TG/HDL-cholesterolCastelli risk index-II (CRI-II) = LDL-cholesterol/HDL-cholesterol.

### 2.3. Echocardiography Evaluation

The left ventricular systolic and diastolic functions were assessed by transthoracic echocardiography. A single investigator performed the resting transthoracic echocardiography using commercially available ultrasound systems (SonoScape SSI-5000 Colour Doppler, SonoScape Medical Corp., Shenzhen, China). Two-dimensional echocardiography techniques, including colour Doppler and pulsed-wave tissue Doppler imaging, were applied. LVEF and LAV were measured using the Simpson’s biplane method. Other data assessing the diastolic function included early and late (A) mitral inflow velocities, deceleration time of the E-wave (DT), isovolumic relaxation time (IVRT), early diastolic velocity assessed at the septal (septal e’) and lateral (lateral e’) sites of the mitral annulus, and E/e’ ratio.

Diastolic dysfunction was identified and graded according to the 2016 EACVI/ASE recommendations, if three or four of the following criteria were met: septal e’ velocity < 7 cm/s, lateral e’ velocity < 10 cm/sec, E/e’ > 14, LAV indexed to body surface (LAVI) > 34 mL/m^2^, and tricuspid regurgitation (TR) velocity > 2.8 m/s. Patients with normal diastolic function had at least two of the following parameters: septal e’ ≥ 8 cm/s, lateral e’ ≥ 10 cm/s and LAVI ≤ 34 mL/m^2^ [7].

### 2.4. Statistical Analyses

Data were analyzed using IBM SPSS Statistics for Windows (version 29, SPSS Inc., Chicago, IL, USA). The descriptive statistical analysis included frequencies, mean and median values, standard deviation, and interquartile ranges. The normality of data distribution was tested using the Kolmogorov–Smirnov test. Parametric *t*-tests and non-parametric Mann–Whitney tests were applied to compare the independent variables between the patients with and without LVDD. Comparisons between patients with normal diastolic function and different grades of diastolic dysfunction were made using the Kruskal–Wallis test for continuous data and the Chi-squared test for categorical variables.

A binary logistic regression model using the enter method was conducted to investigate the risk factors associated with LVDD. The categorical variables found to be associated with LVDD (age groups, associated comorbidities, neuropathy, aortic atheromatosis, the CHA_2_DS_2_-VASc value) were included in the model. The reliability and the goodness of model fit were assessed using the Omnibus test of model coefficients and the Hosmer–Lemeshow test.

The relationship between categorical variables and the DD evolution after 1-year follow-up (worsened vs. stable/improved) was examined using contingency tables. The Chi-squared test for association was used to compare the frequency of categorical variables. Fisher’s exact test was used in a few situations when expected frequencies were less than 5.

Comparisons between different age groups or between patients with stable/improved diastolic function vs. those with deterioration of diastolic function at both evaluations (baseline and 1-year follow-up) were performed using one-way ANOVA or the Kruskal–Wallis test (based on the normality of data distribution and subgroup size). The Wilcoxon signed-ranks test or t-Student test for paired samples was used to compare the change between baseline (T0) and follow-up (T1) for the whole sample and within each age group. Correlation analyses (Pearson and Spearman) were used to identify associations between certain variables. *p*-values were two-tailed with an accepted significance level of 0.05.

## 3. Results

At baseline, 138 participants were selected from the 639 outpatients consecutively presenting for evaluation. The enrollment process and reasons for exclusion (refusal to participate, identification of overt atherosclerotic CVD, ischemic ECG signs, echocardiographic modifications, smoker status, uncontrolled blood pressure, and TG values greater than 400 mg/dL) were outlined in a flowchart developed for a previous publication [36]. After 52 weeks, six subjects were lost to follow-up.

The patients were 57.86 ± 8.82 years old, 49.3% male, had had diabetes for at least 5 years, and had inadequate glycemic control (median HbA_1c_ 7.8%). Among the study population, 31 patients were 65 or older. Comorbidities were noted at baseline in 91.3% of the patients; 75.46% had metabolic dysfunction-associated steatotic liver disease, 71.74% had dyslipidemia, 67.4% had hypertension, and 65.94% had obesity. A high abdominal circumference was present in 97.8% of cases (135 persons). Neuropathy was present in 44.2% of the patients, and diabetic retinopathy in 7.2%.

Changes of antihyperglycemic treatment included the addition of incretin-based drugs (sitagliptin, saxagliptin, or exenatide, the only available options for Romanian patients at the time) to the previous therapy (metformin, sulfonylurea, and/or acarbose) in 92 patients. The evolution of patients between baseline and 52-week visits was described in a previously published paper [36].

All patients had normal systolic function (LVEF 67.14 ± 9.35%). Values of the diastolic parameters were E/A 1.09 ± 0.46, E/e’ 6.54 ± 1.84, EDT 192.88 ± 42.76 ms, IVRT 104.23 ± 18.74 ms and LAVi 43.79 ± 11.84 mL/m^2^. At baseline, 29% had normal diastolic function, 51.4% (71 patients) had grade 1 diastolic dysfunction, 5 had grade 2 DD, 7 had grade 3 DD, and 15 had indeterminate DD. Among people 65 or over, 29 patients had diastolic dysfunction.

The baseline characteristics, including echocardiographic parameters, in the patients with and without LVDD are shown in Table 1.

Compared to the normal diastolic function group, LVDD patients were older (59 vs. 54 years, *p* = 0.002), had a longer diabetes duration (median six years), and approximately 95% presented other comorbidities (*p* = 0.039). Compared to people with normal diastolic function, more LVDD patients presented with diabetic neuropathy (*p* = 0.032). Among the 10 patients with retinopathy, eight had DD (*p* = 0.042).

The BMI, WC and HbA_1c_ values, as well as other insulin resistance parameters, were similar between the two groups. There was also no significant difference concerning the lipid profile, inflammatory status, atherogenic indexes, and kidney function test values.

The echocardiographic assessment of diastolic function at baseline (based on EACVI/ASE recommendations) is detailed in Table 1. Among the echocardiographic parameters, E/e’, IVRT, LAVI, interventricular septum (IVS), and left ventricular posterior wall (LVPW) were significantly higher in patients with diastolic dysfunction (*p* < 0.05), whereas E/A was lower compared to those without LVDD (*p* = 0.027). Although the incidence of mitral annular calcification (MAC) was similar between the two groups, aortic atheromatosis was identified in more patients with LVDD (*p* = 0.039). The mean NT-proBNP level was higher in patients with LVDD, although it did not reach statistical significance.

The Chi-squared test for association revealed that more females than males (59.2% vs. 40.8%, *p* = 0.04) and more patients aged between 50–64 years had grade 1 LVDD (*p* = 0.007). A value of three of the CHA_2_DS_2_-VASc score was observed in 40.8% of patients identified with grade 1 LVDD (*p* = 0.013).

The effects of independent variables on the likelihood of having LVDD were investigated using a binary logistic regression analysis. The age category, comorbidities, and the CHA2DS2-VASc score, neuropathy, and aortic atheromatosis were included in the model using the enter method. The model was statistically significant (*p* < 0.001), explaining between 21.8% (Cox & Snell R-squared) and 31.1% (Nagelkerke R-squared) of the variance in LVDD and correctly classifying 76.1% of cases. The Hosmer–Lemeshow test indicated a good data fit (χ^2^ (7) = 7.321, *p* = 0.396). This multivariate logistic regression analysis identified age over 65 as an independent risk factor for diastolic dysfunction (Exp B = 9.85, 95% CI = 1.29–75.36, *p* = 0.027).

When patients were divided into two groups with and without LVDD based on data found at the 52-week visit, significant age (*p* = 0.001), diabetes duration (*p* = 0.012), uric acid (*p* = 0.021), CHA_2_DS_2_-VASc (*p* = 0.003), and eGFR (*p* = 0.027) differences were found.

After one year, diastolic function remained stable in 95 patients, worsened in 27 patients, and improved in 10. Therefore, 48 patients exhibited normal diastolic function, while grade 1 to 3 LVDD was observed in 50 cases. Figure 1 illustrates the distribution of the study population based on diastolic function at baseline and the 1-year follow-up. The full horizontal bars represent the number of patients at baseline for each stage of diastolic function. The bar divisions indicate the results after a 1-year follow-up.

The prevalence of chronic complications and comorbidities remained stable during the follow-up period.

Table 2 and Table 3 present the evolution of patient data (including echocardiographic parameters) between the baseline and 52-week visits, as well as comparisons between patients distributed in three age groups (<50 years, 50–64 years, and ≥65 years) for both evaluations. Due to the small subgroup size, a non-parametric test for paired data (the Wilcoxon signed-rank test) was performed to compare the median values of the studied variables within each group. Significant differences appear marked in the tables.

The initial values of BMI, WC, FPG, HbA_1c_, lipid profile (except HDL-cholesterol levels), UACR, insulin resistance, and inflammatory markers were homogeneously distributed among patients in the three age groups. Patients over 65 had a significantly lower weight, both at baseline and one year later, compared to the other two groups (T0: 85.40 ± 12.89, *p* = 0.010; T1: 84.37 ± 12.26, *p* = 0.020). As expected, estimated GFR values were significantly lower in the elderly and decreased in all groups one year later (all *p* < 0.01), except for patients < 50 years old. These patients also had the highest HDL-cholesterol levels at both evaluations; however, significant differences were observed only for baseline levels (*p* < 0.001).

Among atherogenic indexes, significant differences were noted at both evaluations for AI and CRI-I, with the elderly exhibiting lower values. A similar trend was noted for AIP and CRI-II; however, the difference was significant only at baseline (*p* < 0.01).

Among inflammatory markers, the highest levels of TNF-alpha were observed in patients older than 65, with statistically significant differences noted at the second evaluation (*p* = 0.004). Interestingly, TNF-alpha levels significantly increased after one year in patients both in the 50 to 64-year-old and over 65-year-old groups (*p* < 0.01).

No significant differences in LVEF, fractional shortening, left ventricle end-systolic and end-diastolic diameters, and IVRT were found between the three age groups, either at baseline or after one year. The elderly patients had the lowest E/A, e’, and lateral s’ values. Meanwhile, E/e’, E-wave deceleration time (EDT), and LAVI were significantly higher in this group (all *p* < 0.01, except for LAVI at baseline). At follow-up, minor modifications in mean e’, EDT and IVRT were observed (*p* < 0.01).

No significant differences were found in the 52-week data between patients with worsened and those with stable or improved diastolic function. The statistical associations between categorical variables and the DD evolution (worsened or stable/improved) after 1-year follow-up are presented in Table 4.

### Correlation Analysis

For baseline data, the correlation analysis identified the following significant weak-to-moderate strength associations in the whole group:E/e’ and: age (r = 0.243, *p* = 0.004), diabetes duration (r = 0.227, *p* = 0.008), BMI (r = 0.190, *p* = 0.027), METS-IR (r = 0.19, *p* = 0.024), CHA_2_DS_2_-VASc score (r = 0.353, *p* < 0.001), TNF-alpha (r = 0.336, *p* = 0.034)E/A and: age (r = −0.386, *p* = 0.003), diabetes duration (r = −0.241, *p* = 0.004), weight (r = 0.319, *p* < 0.001), HDL-cholesterol (r = −0.254, *p* = 0.003), AIP (r = 0.231, *p* = 0.006), CHA_2_DS_2_-VASc score (r = −0.343, *p* <0.001)EDT and: age (r = 0.295, *p* < 0.001), diabetes duration (r = 0.229, *p* = 0.023)LAVI and: age (r = 0.231, *p* = 0.01), HbA_1c_ (r = −0.192, *p* = 0.035), METS-IR (r = 0.193, *p* = 0.035).

Among the LVDD patients, the following associations were found for the baseline data:E/e’ and: age (r = 0.251, *p* = 0.014), diabetes duration (r = 0.302, *p* = 0.008), CHA_2_DS_2_-VASc score (r = 0.345, *p* < 0.001) and METS-IR (r = 0.205, *p* = 0.046)E/A and: age (r = −0.382, *p* < 0.001), diabetes duration (r = −0.267, *p* = 0.008), weight (r = 0.255, *p* = 0.011), HDL-cholesterol (r = −0.259, *p* = 0.010), AIP (r = 0.242, *p* = 0.016)EDT and: age (r = 0.347, *p* < 0.001), diabetes duration (r = 0.229, *p* = 0.023), HbA_1c_ (r = 0.219, *p* = 0.030), HDL-cholesterol (r = 0.204, *p* = 0.044).

In patients without diastolic dysfunction, significant negative associations for the baseline data were identified between: E/e’ and PLR (r = −0.332, *p* = 0.045), E/A and BMI (r = −0.317, *p* = 0.04), and METS-IR (r = 0.376, *p* = 0.017). Additionally, moderate positive correlations were observed between E/A and weight (r = 0.416, *p* = 0.008), waist circumference (r = 0.341, *p* = 0.032), and UACR (r = 0.367, *p* = 0.036).

For the 1-year follow-up data, the following significant associations were observed in the whole group:E/e’ and BMI: (r = 0.182, *p* = 0.037), TNF-alpha (r = 0.294, *p* = 0.01), CHA_2_DS_2_-VASc score (r = 0.300, *p* < 0.001)E/A and: age (r = −0.429, *p* < 0.001), diabetes duration (r = −0.277, *p* = 0.009), weight (r = 0.314, *p* = 0.008), CHA_2_DS_2_-VASc score (r = −0.381, *p* < 0.001), TNF-alpha (r = −0.210, *p* = 0.016), CRI-I (r = 0.192, *p* = 0.027), CRI-II (r = 0.192, *p* = 0.027), AI (r = 0.192, *p* = 0.027)EDT and: age (r = 0.385, *p* < 0.001), diabetes duration (r = 0.227, *p* = 0.01)IVRT and: diabetes duration (r = 0.240, *p* = 0.006), hsCRP (r = −0.913, *p* = 0.028), index C-peptide (r = −0.179, *p* = 0.040)LAVI and: age (r = 0.288, *p* < 0.001), HDL-cholesterol (r = 0.223, *p* = 0.043).

Among the LVDD patients, E/A correlated with age (r = −0.447, *p* < 0.001), diabetes duration (r = −0.301, *p* = 0.006), weight (r = 0.262, *p* = 0.017), and CHA_2_DS_2_-VASc score (r = −0.324, *p* = 0.002). Additionally, positive associations were noted between: EDT and age and diabetes duration (r = 0.416, *p* < 0.001 and r = 0.290, *p* = 0.008, respectively), IVRT and HbA_1c_ (r = 0.237, *p* = 0.036), E/e’ and TNF-alpha (r = 0.260, *p* = 0.02).

Among patients with normal diastolic function, significant positive associations were observed between: LAVI and weight (r = 0.360, *p* = 0.011) and uric acid (r = 0.307, *p* = 0.038), E/e’ and CHA_2_DS_2_-VASc score (r = 0.525, *p* < 0.001), E/A and weight (r = 0.343, *p* = 0.014). Additionally, negative associations were noted between the CHA_2_DS_2_-VASc score and E/A, as well as LAVI (*p* < 0.05).

Results of the correlation analysis between metabolic and inflammatory markers and diastolic function parameters are presented in Table 5 (baseline data) and Table 6 (1-year follow-up data).

## 4. Discussion

In this study of 138 adult T2DM patients with poor glycemic control and no atherosclerotic CVD, LVDD was found in 71% of cases using conventional echocardiography and tissue Doppler imaging. Grade 1 LVDD was observed in 71 patients. Compared to other reports in the literature, where LVDD ranged from 14.4% to 78.89%, the prevalence in our study is at the higher end [9,37,38,39,40,41,42,43]. Hoek et al. systematically reviewed 65 studies of moderate to high heterogeneity which used a ≥50% cut-off value for LVEF and were published between 2016 and 2022. Based on a sensitivity analysis, they reported a 43% prevalence of LVDD among 25,729 in- and outpatients, with grade 1 LVDD most commonly met [9].

Asymptomatic LVDD and HFpEF are more frequent among T2DM patients than heart failure with reduced ejection fraction (HFrEF), heart failure with mildly reduced ejection fraction (HFmrEF), and left ventricular systolic dysfunction (LVSD). Diastolic dysfunction is an early manifestation of diabetic cardiomyopathy, wherein structural and functional myocardial dysfunction may occur in the absence of conventional cardiovascular diseases. Early changes brought by LVDD include delayed left ventricular filling, often without an impaired cardiac systolic function [44].

Age is linked to LVDD and serves as an independent predictor of diastolic function decline over time. Other risk factors for diastolic dysfunction that literature reports with low to moderate strength of evidence include diabetes duration, BMI, hypertension, dyslipidemia, retinopathy, gender, glycemic control (HbA_1c_, but also glycemic variability), insulin therapy, atherogenic indices and some biomarkers, such as asprosin [10,44,45].

The patients enrolled in this study had an average age of 57.86 years and a diabetes history of 5 years; 29 patients in the elderly group had LVDD. Most patients also had other metabolic conditions, such as excess weight, dyslipidemia, hypertension, and metabolic fatty liver disease. Grade 1 LVDD was associated with female sex (*p* = 0.04) and a CHA_2_DS_2_-VASc value of 3 (*p* = 0.013). Patients with LVDD were older, had a longer diabetes duration (a 6-year median), other associated chronic diseases, and neuropathy as the most constant chronic complication of diabetes. Their metabolic profile (including calculated atherogenic indexes), eGFR and inflammatory status were similar to those of patients with normal diastolic function. As age is a non-modifiable factor, and ageing influences the physiological processes that contribute to diastolic decline, it is essential to adopt a multi-pillar treatment approach focusing on glycemic control, lipid profile, weight control, and blood pressure to improve cardiometabolic outcomes. Clinicians should integrate lifestyle interventions (exercise, diet) and antihyperglycemic therapies with cardiovascular benefits to counteract the deleterious effects of age on the cardiac structure and function [22,23].

Unlike heart failure markers such as NT-proBNP, serum biomarkers to identify LVDD in its early stages are not yet conventionally available [46]. A meta-analysis reported low sensitivity and specificity levels of the natriuretic peptides for detecting LVDD and HFpEF. Natriuretic peptides are elevated in older individuals and those with impaired kidney function, and are inversely related to BMI values. Under these circumstances, they can rule out LVDD or HFpEF but cannot diagnose these conditions [13]. The LVDD patients in our study had a higher level of NT-proBNP, but this difference was not statistically significant.

In a study on 55 T2DM Greek patients, the only significant relationship found by logistic regression analysis was a positive correlation between LVDD and age (*p* < 0.001) [47].

In a case–control study enrolling hospitalized T2DM patients in Egypt, the grade 2 and 3 LVDD group exhibited significantly higher levels of serum creatinine, hs-CRP, uric acid, and NLR than patients with mild or no LVDD. Among these markers, hs-CRP predicted grade 2 or higher LVDD in both univariate and multivariate models (OR 16.5, 95% CI: 1.7–157.5, *p* = 0.015, and OR 17.5, 95% CI: 1.6–184.6, *p* = 0.018) [48]. In our study, patients with LVDD had higher levels of hs-CRP as well as IL-6 and TNF-α, but no significant differences were noted when compared to patients without LVDD. The hs-CRP level in the elderly patients was lower than in the other two age groups; however, this difference was also statistically insignificant at both evaluations.

Yang et al. reported a 47.8% prevalence of LVDD in 855 hospitalized T2DM patients in China. LVDD prevalence was increased (56.77%) in groups with higher levels of neutrophil-to-lymphocyte ratio (NLR). However, multivariate regression analysis demonstrated that patients’ age, systolic blood pressure, and creatinine levels were significantly associated with LVDD, whereas NLR, diabetes duration, and HbA_1c_ were not (all *p*-values > 0.05). Even so, the authors suggested that NLR, a marker of low-grade chronic inflammation, could be indicative of early impairment of diastolic function, and they therefore proposed it should be used to screen elderly patients with T2DM, hypertension, and declining renal function [15]. In our study, we noticed no significant differences in NLR between patients with or without LVDD.

In a cross-sectional study of 203 otherwise healthy T2DM patients from the Democratic Republic of Congo, 47.8% of patients had LVDD according to the EACVI/ASE 2016 diastolic function assessment recommendations. Compared to those with normal diastolic function, LVDD patients (mean age 63.9 ± 10.5 years) were more likely to be over 55 (*p* < 0.001) and have obesity, dyslipidaemia, elevated LDL-cholesterol (*p* = 0.028), and significantly higher levels of AIP and CRI-II (*p* < 0.05). Multivariable analysis identified high AIP (OR 3.37) and CRII-II (OR 3.8) as independent determinants of LVDD [14]. Our elderly patients had lower values for AI, CRI-I, CRI-II, and AIP; however, we did not identify significant differences between patients with and without LVDD.

A case–control study from Ethiopia, involving 100 T2DM patients matched with 200 normotensive individuals without diabetes, all aged over 60, found that dyslipidemia, metformin and glibenclamide treatment, high serum triglyceride levels, the presence of neuropathy, and the use of statins were significantly correlated with the presence of LVDD [49].

In a study conducted in India, the prevalence of LVDD was 100% in patients over 75 years and in those with diabetes for more than 20 years, whereas the total prevalence was 78.89%. Grade 1 DD was found in 41.11% of cases and was more commonly met in patients with HbA_1c_ levels greater than 10% [38]. The odds ratio for the elderly patients in our study to have LVDD was 9.85. They had the lowest E/A, e’, and lateral s’ values, and significantly higher E/e’, EDT, and LAVI values. At follow-up, minor modifications in mean e’, EDT and IVRT were observed (*p* < 0.01). Significant differences between patients with and without DD at the 1-year follow-up were noted for uric acid, CHA_2_DS_2_-VASc score, and eGFR.

In a small study from Eastern India, LVDD prevalence among newly diagnosed, asymptomatic, non-hypertensive T2DM patients was 53%. Age and HbA_1c_ were identified as independent predictors for LVDD. The cut-off value for HbA_1c_ was 9.5%, with a positive predictive value of 91.62% for detecting LVDD. An adjusted OR of 2.625 (CI: 1.264–5.450, *p* = 0.010) indicated that each 1% increase in the HbA_1c_ levels amplified the risk of LVDD by 2.6 [50].

In a cross-sectional study conducted on 114 T2DM Chinese inpatients with preserved ejection fraction, a subgroup analysis indicated that a longer diabetes duration (>2 years) and poorly controlled blood glucose levels (HbA_1c_ > 7.5%) were linked to an increased severity of diastolic dysfunction. Multiple linear regression analysis identified HbA_1c_ and disease duration as independent risk factors for low E/A and elevated E/e’ (*p* < 0.001) [44].

A study on asymptomatic T2DM Japanese patients with preserved LVEF and no coronary artery disease found that E/e’ was significantly higher among patients with high glucose variability (≥35.9 mg/dL) than in those with low glucose variability (11.3 ± 3.9 vs. 9.8 ± 2.8, *p* = 0.03), even though age and HbA_1c_ were similar in the two groups [51].

Given these differing results across existing studies, it can be said that the evidence supporting the hypothesized association between HbA_1c_ and diastolic parameters remains, for now, inconclusive. Guria et al. reported that higher HbA_1c_ levels increased the risk of LVDD (OR 1.26), whereas other researchers did not find similar results in overweight T2DM patients, potentially suggesting that BMI may be a confounding factor [51,52].

In our study, HbA_1c_ correlated only with EDT in LVDD patients at baseline and with IVRT after one year. While age and diabetes duration correlated with nearly all diastolic parameters in the entire sample, only diabetes duration maintained the associations with E/A and EDT in the LVDD group at both evaluation times. We observed minimal but significant increases of 0.14 for E/e’ and 0.3 for lateral s’ among all patients, particularly within the 54–60 years age group. Furthermore, the mean values for LVEF, LAVI, and E/e’ showed slight decreases in time, while e’, IVRT, and EDT increased significantly, especially in the elderly patients.

A single-centre observational study conducted in South India during the coronavirus pandemic concluded that half of the 80 T2DM inpatients who had neither CVD nor hypertension exhibited LVDD. In this regard, diabetes duration made a difference, ranging from 15 years in patients with LVDD to 11 years in the absence of LVDD. LVDD was also associated with neuropathy and microvascular chronic complications (*p* < 0.005), as well as higher levels of HbA_1c_ (9.5% vs. 8.2%, *p* = 0.003). In this case, multivariate logistic regression analysis identified retinopathy as an independent and predictive factor for LVDD (OR 9.11, *p* = 0.001) [53].

Longitudinal research on the long-term impact of T2DM on diastolic function is limited to only a few studies and relatively small patient samples. In a 7-year follow-up study of asymptomatic T2DM patients without CVD, LVDD incidence increased from 53% to 61% (*p* = 0.004) compared to a control group without diabetes [54]. In 310 T2DM patients without overt CVD, Bergerot et al. identified age, retinopathy, and higher systolic blood pressure as independent parameters associated with the deterioration of diastolic function after 3 years [55]. In a 1-year interventional study, septal e’, E/e’, and global longitudinal strain showed the most improvement in patients with uncontrolled diabetes and normal LVEF, whose HbA_1c_ levels decreased most substantially [56]. In our study, a comparative analysis between patients with worsened and those with stable or improved diastolic function showed no significant differences after 52 weeks. As we have previously reported, even though significant yet modest changes in LV end diastolic and systolic diameters and IVRT were observed in patients on incretin-based medication for 52 weeks, no significant differences between incretin and non-incretin therapies were identified [8].

Our results highlight inconsistencies in the literature. Glycaemic control, diabetes duration, obesity, dyslipidaemia and hypertension were not independently associated with asymptomatic left ventricular diastolic dysfunction (LVDD), which is consistent with some studies but inconsistent with others [43,57,58].

Methodologically, developing a nomogram could facilitate the recognition of risk factors that contribute to the development and progression of LVDD, as well as the identification of potentially reversible causes, such as stricter glycemic control and weight reduction. An example in this direction is the study by Xia et al., conducted on 63 patients with metabolic liver disease, of whom 51 had diabetes. The authors found that the number of comorbidities (T2DM, obesity, and hypertension), HbA_1c_ levels, and the epicardial adipose tissue volume index, as assessed by cardiac magnetic resonance, were independently associated with LVDD (all *p* < 0.05). The authors included these independent factors in a nomogram that can be used for screening [59].

### Limitations

The advantage of consecutive patient enrollment based on strictly relevant inclusion and exclusion criteria was partially counterbalanced by the smaller patient numbers in some study subgroups, which limits the added value brought by multivariate analyses and adjustments for confounders such as age and gender. In our study, we did not assess glycemic and blood pressure variability, hypertension severity and duration, including related treatments, which have been identified in others’ research as predictive of diastolic dysfunction. The small number of cases with specific complications, such as diabetic retinopathy, did not allow their inclusion in the statistical analysis as a possible risk factor for LVDD progression.

Moreover, as this is a post hoc observational analysis, it cannot establish causality; enrollment in multiple centres and a longer follow-up period would have enabled the collection of more patient data and a comprehensive assessment of factors predicting LVDD, including diabetes duration, specific medications, glycemic control thresholds, and treatment of other modifiable risk factors. These measures would also have avoided type I and type II errors, which are potentially possible in small subgroup analyses.

Regarding the inflammatory markers, our findings showed contradictory patterns, with decreased hsCRP but increased IL-6 and TNF-α after one year, particularly in elderly patients. Despite metabolic improvement, chronic subclinical inflammation persisted. Undiagnosed, subclinical arthropathy could explain elevated TNF-α levels despite no reported joint pain. Our patients received the antihyperglycemic drugs available at the time of enrollment, rather than the newer drugs now known to provide cardiovascular benefits; therefore, a comparison between medication classes is not possible with this dataset alone. Finally, the study design did not include the dietary habits and physical activity levels of these outpatients.

Although these limitations restrict broad generalization, they do not detract from the clinical practice-targeting message that assessment of the diastolic and systolic function is also important for diabetes patients without atherosclerotic manifestations and regardless of the following characteristics: glycemic control, insulin resistance, and inflammatory status. This is especially useful for patients over the age of 65 years, who often have diabetic neuropathy and other comorbidities.

Therefore, it is essential to continue analyzing information related to diastolic dysfunction dynamics to identify the correct strategy for determining the risk factors associated with diastolic dysfunction in T2DM patients. This will help clinicians to prevent progression to heart failure. Cardiovascular diseases remain the primary cause of mortality among patients with T2DM, even though diabetes-related therapies have known significant advancements in recent years, when drugs with proven cardio-renal-metabolic benefits were introduced in clinical practice. In the 11th edition of the *IDF Atlas*, the high risk of cardiovascular diseases in T2DM people corresponds to an 84% higher risk of heart failure, a 72% higher risk of heart attack and a 52% higher risk of stroke [60]. As to the prevalence of diabetes in people aged 65–99 years, if the actual trend continues, the projection to 2050 is that 278 million people over 65 years will have diabetes, with a significant impact on public health and economic challenges [60].

## 5. Conclusions

Asymptomatic left ventricular diastolic dysfunction is common in type 2 diabetes patients without atherosclerotic manifestations. Patients with LVDD have a longer diabetes duration and a higher prevalence of diabetic neuropathy. Age is the only significant risk factor for diastolic dysfunction. Almost all elderly patients have LVDD. After a 1-year follow-up, patients with worsening diastolic function have similar features to those with stable or improved diastolic function. Further research is necessary to identify the full range of factors associated with the deterioration of diastolic function over time, thereby helping to characterize at-risk patients and facilitating the prevention of heart failure progression.

## Figures and Tables

**Figure 1 jcm-14-05772-f001:**
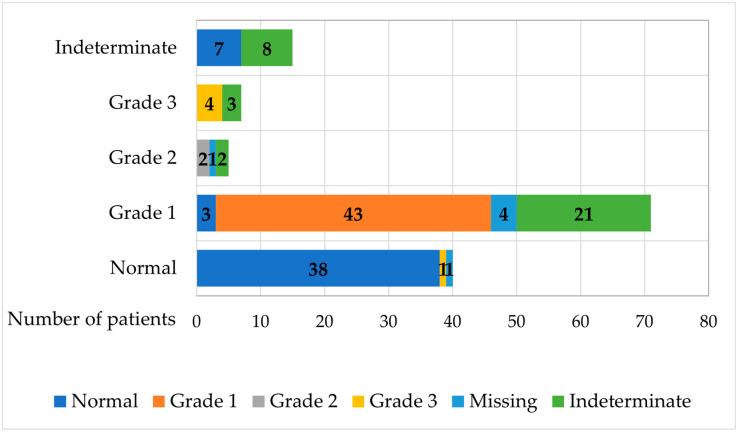
Evolution of diastolic function during the 1-year follow-up.

**Table 1 jcm-14-05772-t001:** Patients’ characteristics at baseline, overall and in the groups with and without LVDD.

Studied Variable	Overall (N = 138)	With DD (N = 98)	Without DD (N = 40)	*p*-Value
Age (years)	57.86 ± 8.82	59.24 ± 8.77	54.45 ± 8.05	0.002 **
<50 (%)	21.7	15.3	37.5	<0.001 **
50–64 (%)	55.8	55.1	57.5	
>65 (%)	22.5	29.6	5	
Sex (male%)	49.30	46.9	55	0.390
BMI (kg/m^2^)	32.65 ± 5.50	32.88 ± 5.74	32.07 ± 4.88	0.678
Waist circumference (cm)	109.13 ± 10.74	109.22 ± 11.03	109.39 ± 10.18	0.849
Diabetes duration * (years)	5 (8)	6 (8)	3 (9)	0.122
Neuropathy (%)	44.2	50	30	0.032 **
HbA_1c_ * (%)	7.8 (1.11)	7.8 (1.03)	7.8 (1.65)	0.562
Fasting glycemia * (mg/dL)	163.5 (46)	165.5 (51)	162 (39)	0.835
Insulin * (µIU/mL)	11.2 (9.39)	11.95 (11.2)	12.60 (6.72)	0.882
C-peptide * (ng/mL)	3.26 (2.22)	3.18 (1.9)	2.89 (1.92)	0.678
HOMA-IR	5.74 ± 3.87	5.76 ± 3.97	5.69 ± 3.68	0.932
HOMA C-peptide	4.02 ± 2.10	4.01 ± 2.12	4.06 ± 2.10	0.914
Index C-peptide *	0.24 (0.19)	0.23 (0.17)	0.26 (0.23)	0.938
Total cholesterol (mg/dL)	195.33 ± 46.11	196.29 ± 44.68	193.00 ± 49.96	0.706
LDL-cholesterol (mg/dL)	103.12 ± 38.96	102.81 ± 40.02	103.90 ± 36.70	0.882
HDL-cholesterol (mg/dL)	56.79 ± 15.27	58.26 ± 15.90	53.19 ± 13.07	0.104
Non-HDL-cholesterol (mg/dL)	138.53 ± 44.88	138.05 ± 43.42	139.80 ± 48.84	0.832
Triglycerides (mg/dL)	202.57 ± 90.46	203.57 ± 88.25	199.70 ± 96.78	0.683
TyGi	5.14 ± 0.25	5.17 ± 0.26	5.14 ± 0.25	0.632
AIP	0.165 ± 0.25	0.159 ± 0.25	0.178 ± 0.24	0.686
AI	2.62 ± 1.11	2.56 ± 1.09	2.77 ± 1.17	0.310
LCI	20.20 (23.15)	19.72 (22.16)	20.95 (29.25)	0.618
CRI-I	3.62 ± 1.11	3.55 ± 1.09	3.77 ± 1.17	0.310
CRI-II	1.94 ± 0.90	1.89 ± 0.92	2.06 ± 0.86	0.327
METS-IR	51.37 ± 9.65	50.88 ± 8.21	51.80 ± 8.93	0.851
eGFR (mL/min/1.73 m^2^)	83.09 ± 18.00	82.41 ± 18.40	84.78 ± 17.08	0.243
UACR (mg/g)	27.14 ± 48.64	23.18 ± 36.02	35.04 ± 67.17	0.859
Uric acid (mg/dL)	5.48 ± 1.43	5.54 ± 1.47	5.33 ± 1.31	0.429
hsCRP * (mg/L)	5.35 (9.18)	5.36 (8.91)	5.29 (11.4)	0.747
IL-6 (pg/mL)	3.52 ± 4.66	3.29 ± 2.72	2.99 ± 1.69	0.533
TNF-α (pg/mL)	8.05 ± 3.85	8.08 ± 3.87	7.97 ± 3.86	0.814
NLR *	1.85 (1.01)	1.81 (0.96)	1.91 (1.1)	0.927
PLR *	107.63 (49.66)	107 (48.43)	109.13 (73.6)	0.937
With comorbidities (%)	91.3	94.9	82.5	0.039 **
Obesity (%)	65.94	64.3	70	0.735
Dyslipidemia (%)	71.74	75.5	62.5	0.146
Hypertension (%)	67.4	71.4	57.5	0.161
Steatohepatitis (%)	75.46	78.6	72.5	0.673
LVEF (%)	67.14 ± 9.35	66.43 ± 9.44	68.88 ± 9.00	0.164
FS (%)	38.16 ± 7.89	37.63 ± 7.90	39.38 ± 7.82	0.249
E/A *	0.95 (0.69)	0.86 (0.61)	1.22 (0.75)	0.027 **
E/e’	6.54 ± 1.84	6.94 ± 1.86	5.60 ± 1.41	<0.001 **
EDT * (ms)	190 (60)	188 (65)	190 (45)	0.590
IVRT * (ms)	105 (25)	105 (30)	100 (24)	0.044 **
IVS * (mm)	11 (2)	12 (2.5)	11 (2)	0.006 **
LVPW * (mm)	11 (2)	11 (2)	10.5 (2)	0.048 **
LAVI * (mL/m^2^)	42 (18)	43.5 (17)	35.5 (11)	0.001 **
CHA_2_DS_2_-VASc score *	3 (1)	3 (1)	2 (1)	0.002 **
NT-proBNP * (pg/mL)	63 (77.1)	65 (87)	53 (77)	0.377
MAC (%)	43.47	45.9	37.5	0.450
AA (%)	11.6	15.3	2.5	0.039 **

* Data are expressed as medians and IQR (abnormal distribution); ** Statistical significance; DD: diastolic dysfunction; BMI: body mass index; HbA_1c_: glycated hemoglobin; HOMA-IR: Homeostatic Model Assessment of Insulin Resistance; HOMA C-peptide: Homeostatic Assessment Model of C-peptide; TyGi: Triglyceride Glucose Index; AIP: atherogenic index of plasma; AI: atherogenic index; LCI: lipoprotein combine index; CRI-I: Castelli risk index I; CRI-II: Castelli risk index II; METS-IR Metabolic Score for Insulin Resistance; eGFR: estimated glomerular filtration rate; UACR: urinary albumin-to-creatinine ratio; hsCRP: high-sensitivity C-reactive protein; IL-6: interleukin 6; TNF-α: tumour necrosis factor-alpha; NLR: neutrophil-to-lymphocyte ratio; PLR: platelet-to-lymphocyte ratio; LVEF: left ventricular ejection fraction; FS: fractional shortening; A: mitral A wave velocity (atrial contraction) with pulsed Doppler; E: mitral E wave velocity (rapid filling) with pulsed Doppler; e’: mitral annular velocity with tissue Doppler imaging; DT: E-wave deceleration time; IVRT: isovolumic relaxation time; IVS: interventricular septum; LVPW: left ventricular posterior wall; LAVI: indexed left atrium volume; NT-proBNP: N terminal-brain natriuretic peptide; MAC: mitral annular calcification; AA: aortic atheromatosis.

**Table 2 jcm-14-05772-t002:** Evolution of metabolic parameters between baseline and 52-week follow-up, overall and in the age-based groups.

	Overall		Age Groups		
Variables		<50	50–64	>65	*p*-Values
Number T0	138	30 (21.7%)	77 (55.8%)	31 (22.5%)	
Number T1	132	29 (21.7%)	73 (55.3%)	30 (22.73%)	
Diabetes duration (years) *	5 (8)	1 (2.25)	2 (5)	6 (4)	<0.001 **
BMI T0	32.65 ± 5.50	33.28 ± 6.98	32.74 ± 4.94	31.82 ± 5.32	0.519
BMI T1	31.97 ± 5.00	32.10 ± 5.00	32.14 ± 4.90	31.41 ± 4.85	0.722
*p*-values (T1 − T0)	<0.001 **	0.006 **	0.019 **	0.023 **	
HbA_1c_ * (%) T0 HbA_1c_ * (%) T1	7.8 (1.11)	7.8	7.8	7.8	0.728
	7.2 (1.1)	7.2	7.4	7.1	0.526
*p*-values (T1 − T0)	<0.001 **	0.004 **	<0.001 **	0.002 **	
Fasting glycemia * (mg/dL) T0 Fasting glycemia * (mg/dL) T1 *p*-values (T1 − T0)	163.5 (46)	170 (44)	159 (42)	166 (62)	0.434
	145 (40.5)	142 (39)	148 (50.5)	147 (39.25)	0.528
	<0.001 **	<0.001 **	0.025 **	0.020 **	
eGFR (mL/min/1.73 m^2^) T0	83.09 ± 18.00	95.80 ± 14.65	83.61 ± 16.36	69.52 ± 15.53	<0.001 **
eGFR (mL/min/1.73 m^2^) T1	77.25 ± 17.66	91.96 ± 16.68	77.27 ± 14.27	63.52 ± 15.23	<0.001 **
*p*-values (T1 − T0)	<0.001 **	0.238	0.003 **	0.002 **	
UACR (mg/g) T0	27.14 ± 48.64	18.74 ± 22.00	33.26 ± 60.57	18.18 ± 22.28	0.600
UACR (mg/g) T1	17.30 ± 29.80	17.26 ± 30.42	19.70 ± 34.81	14.41 ± 7.11	0.763
*p*-values (T1 − T0)	0.016 **	0.587	0.014 **	0.546	
hsCRP * (mg/L) T0	5.35 (9.18)	5.47 (11.75)	5.33 (14.68)	4.15 (6.43)	0.511
hsCRP * (mg/L) T1	3.61 (6.56)	4.04 (7.97)	3.58 (6.64)	3.14 (6.16)	0.849
*p*-values (T1 − T0)	<0.001 **	0.031 **	0.006 **	0.236	
IL-6 (pg/mL) T0	3.52 ± 4.66	2.58 ± 1.00	3.40 ± 2.91	3.28 ± 2.19	0.229
IL-6 (pg/mL) T1	3.79 ± 2.43	3.42 ± 2.05	3.95 ± 2.71	3.74 ± 2.05	0.568
*p*-values (T1 − T0)	0.033 **	0.508	0.040 **	0.629	
TNF-α (pg/mL) T0	8.05 ± 3.85	6.89 ± 2.31	8.07 ± 4.02	9.13 ± 4.38	0.067
TNF-α (pg/mL) T1	9.34 ± 5.98	7.22 ± 3.00	9.21 ± 4.28	11.69 ± 9.86	0.004 **
*p*-values (T1 − T0)	<0.001 **	0.689	0.009 **	0.014 **	

* Data are expressed as medians and IQR (abnormal distribution); ** Statistical significance; T0: baseline; T1: 1-year follow-up; BMI: body mass index; HbA_1c_: glycated hemoglobin; eGFR: estimated glomerular filtration rate; UACR: urinary albumin-to-creatinine ratio; hsCRP: high-sensitivity C-reactive protein; IL-6: interleukin 6; TNF-α: tumour necrosis factor-alpha.

**Table 3 jcm-14-05772-t003:** Evolution of echocardiographic data between baseline and 52-week follow-up, overall and in the age-based groups.

	Overall		Age Groups		
Variables		<50	50–64	>65	*p*-Values
LVEF (%) T0	67.14 ± 9.35	67.40 ± 8.79	67.40 ± 8.79	67.40 ± 8.79	0.651
LVEF (%) T1	66.66 ± 7.13	67.21 ± 5.80	66.45 ± 7.03	66.67 ± 8.55	0.842
*p*-values (T1 − T0)	0.213	0.665	0.551	0.248	
FS % T0	38.16 ± 7.89	38.93 ± 7.19	37.64 ± 8.42	38.74 ± 7.23	0.619
FS % T0	37.89 ± 6.97	38.93 ± 7.21	37.29 ± 6.74	38.37 ± 7.36	0.619
*p*-values (T1 − T0)	0.337	0.713	0.470	0.500	
LVEDD (mm) * T0	48 (10)	48.5 (9)	48 (10)	47 (9)	0.606
LVEDD (mm) * T1	50 (10)	50 (10.5)	50 (10.5)	48 (9)	0.529
*p*-values (T1 − T0)	0.022 **	0.965	0.019 **	0.190	
LVESD (mm) T0	28.44 ± 7.97	29.53 ± 6.57	29.09 ± 8.15	25.87 ± 7.68	0.107
LVESD (mm) T1	29.47 ± 8.06	28.96 ± 8.47	30.63 ± 8.42	27.17 ± 6.33	0.133
*p*-values (T1 − T0)	0.076	0.374	0.060	0.056	
E/A * T0	0.95 (0.69)	1.39 (0.47)	0.90 (0.66)	0.73 (0.33)	<0.001 **
E/A * T1	0.94 (0.76)	1.47 (0.47)	0.94 (0.73)	0.76 (0.34)	<0.001 **
*p*-values (T1 − T0)	0.792	0.779	0.793	0.614	
E/e’ T0	6.54 ± 1.84	6.06 ± 1.65	6.39 ± 1.68	7.38 ± 2.14	0.014 **
E/e’ T1	6.68 ± 1.80	6.22 ± 2.15	6.80 ± 1.55	6.84 ± 1.97	0.081
*p*-values (T1 − T0)	0.034 **	0.393	<0.001 **	0.156	
Mean e’ (cm/s) T0	8.21 ± 1.87	9.66 ± 2.19	8.04 ± 1.47	7.23 ± 1.61	<0.001 **
Mean e’ (cm/s) T1	8.25 ± 1.83	9.65 ± 2.05	8.00 ± 1.54	7.51 ± 1.58	<0.001 **
*p*-values (T1 − T0)	0.453	0.980	0.783	0.011 **	
Lateral s’ * (cm/s) T0	6.7 (1.6)	7 (1.9)	6.85 (1.4)	6 (1)	0.003 **
Lateral s’ * (cm/s) T1	7 (1.8)	7.5 (2.3)	7.1 (1.8)	6.05 (1.6)	0.002 **
*p*-values (T1 − T0)	<0.001 **	0.103	0.014 **	0.109	
EDT * (ms) T0	190 (60)	172.5 (63.25)	190 (62.5)	200 (55)	0.004 **
EDT * (ms) T1	200 (57.5)	165 (53.75)	200 (60)	215 (57.5)	<0.001 **
*p*-values (T1 − T0)	0.002 **	0.548	0.046 **	0.004 **	
IVRT * (ms) T0	105 (25)	100 (16.25)	105 (25)	100 (30)	0.218
IVRT * (ms) T1	105 (20)	100 (25)	110 (20)	105 (35)	0.149
*p*-values (T1 − T0)	0.002 **	0.468	0.078	0.002 **	
IVS * (mm) T0	11 (2)	11 (2)	11.5 (2)	12 (3)	0.246
IVS * (mm) T1	12 (2)	11 (2)	12 (3)	12 (2.12)	0.047 **
*p*-values (T1 − T0)	0.186	0.588	0.131	0.487	
LVPW * (mm) T0	11 (2)	11 (2)	11 (2)	11 (2)	0.650
LVPW * (mm) T1	11 (2)	11 (2)	11 (2)	11 (2)	0.104
*p*-values (T1 − T0)	0.039 **	0.265	0.040 **	0.065	
LAVI * (mL/m^2^) T0	42 (18)	36 (13.5)	42 (18)	45 (15)	0.050
LAVI * (mL/m^2^) T1	42	38 (10.5)	43 (14.75)	44.5 (12.5)	0.003 **
*p*-values (T1 − T0)	0.133	0.955	0.333	0.194	
LA area (cm^2^)	24.36 ± 4.28	22.73 ± 3.44	24.71 ± 4.52	25.03 ± 4.13	0.025 **
LA area (cm^2^)	24.17 ± 4.22	22.72 ± 3.76	24.42 ± 4.20	24.97 ± 4.84	0.050
*p*-values (T1 − T0)	0.882	0.773	0.889	0.742	

* Data are expressed as medians and IQR (abnormal distribution); ** Statistical significance; T0: baseline; T1: 1-year follow-up; LVEF: left ventricular ejection fraction; FS: fractional shortening; LVEDD: LV end-diastolic diameter; LVESD: LV end-systolic diameter; A: mitral A wave velocity (atrial contraction) with pulsed Doppler; E: mitral E wave velocity (rapid filling) with pulsed Doppler; e’: mitral annular velocity with tissue Doppler imaging; EDT: E-wave deceleration time; IVRT: isovolumic relaxation time; IVS: interventricular septum; LVPW: left ventricular posterior wall; LAVI: indexed left atrium volume; LA: left atrium.

**Table 4 jcm-14-05772-t004:** Associations between the studied variables and the evolution of diastolic function.

Variables		Worsened		Stable/Improved		Total		*p*-Value
		N (27)	%	N (105)	%	N (132)	%	
Gender	Male	16	59.3	49	46.7	65	49.2	0.243
	Female	11	40.7	56	53.3	67	50.8	
Age	<50	5	18.5	24	22.9	29	22.0	0.845
	50–64	15	55.6	58	55.2	73	55.3	
	>65	7	25.9	23	21.9	30	22.7	
Diabetes duration (years)	<5	14	51.9	46	43.8	60	45.5	0.561
	5–10	8	29.6	29	27.6	37	28.0	
	>10	5	18.5	30	28.6	35	26.5	
Comorbidities	With	25	92.6	97	92.4	122	92.4	1.000
	Without	2	7.4	8	7.6	10	7.6	
Obesity	With	15	55.6	74	70.5	89	67.4	0.140
	Without	12	44.4	31	29.5	43	32.6	
Dyslipidemia	With	21	77.8	75	71.4	96	72.7	0.509
	Without	6	22.2	30	28.6	36	27.3	
Hypertension	With	17	63	74	70.5	91	68.9	0.452
	Without	10	37	31	29.5	41	31.1	
Steatohepatitis	With	24	88.9	73	69.5	97	73.5	0.051
	Without	3	11.1	32	30.5	35	26.5	
Neuropathy	With	11	40.7	46	43.8	57	43.2	0.774
	Without	16	59.3	59	56.2	75	56.8	
MAC	With	12	44.4	45	42.9	57	43.2	0.882
	Without	15	55.6	60	57.1	75	56.8	
HbA_1c_ (%)	<7.5	9	33.3	37	35.2	46	34.8	0.365
	7.5–8	11	40.7	29	27.6	40	30.3	
	>8	7	25.9	39	37.1	46	34.8	
hs CRP (mg/L)	<3	8	29.6	29	28.4	37	28.7	0.903
	>3	19	70.4	73	71.6	92	71.3	
HOMA-IR	<5	12	44.4	55	52.4	67	50.8	0.462
	>5	15	55.6	50	47.6	65	49.2	
Antihyperglycemic drugs	Incretin-based	22	81.5	67	63.8	89	67.4	0.081
	Conventional	5	18.5	38	36.2	43	32.6	

N: number; MAC: mitral annular calcification; HbA_1c_: glycated hemoglobin; hsCRP: high-sensitivity C-reactive protein; HOMA-IR: Homeostatic Model Assessment of Insulin Resistance. Incretin-based treatment included: sitagliptin, saxagliptin or exenatide add-on to previous drugs. Conventional treatment included: metformin and/or sulfonylurea or acarbose.

**Table 5 jcm-14-05772-t005:** Associations between the echocardiographic parameters and other patient characteristics at baseline, overall and in the groups with and without LVDD.

LVDD	Echocardiographic Parameters	Age	Diabetes Duration	BMI	WC	Weight	CHA_2_DS_2_-VASc Score	HbA_1c_	HDL- Cholesterol	METS -IR	AIP	UACR	TNF-α	hs CRP	PLR
Overall		0.243 **	0.227 **	0.190 *	0.139	0.053	0.353 **	−0.017	0.089	0.190 *	−0.002	0.072	0.336 *	−0.103	−0.061
With DD	E/e’	0.251 *	0.302 **	0.167	0.154	0.053	0.345 **	−0.184	0.069	0.205 *	−0.014	0.164	0.097	−0.095	−0.005
Without DD		0.033	−0.037	0.207	0.134	0.013	0.217	−0.100	−0.067	0.232	0.098	0.065	0.336 *	−0.083	−0.332 *
Overall		−0.386 **	−0.241 **	0.061	0.065	0.319 **	−0.343 **	−0.158	−0.254 **	0.163	0.231 **	0.170	−0.072	0.036	0.042
With DD	E/A	−0.382 **	−0.267 **	−0.006	−0.015	0.255 *	−0.303 **	−0.161	−0.259 *	0.096	0.242 *	0.048	−0.116	0.004	0.069
Without DD		−0.237	−0.063	−0.317 *	0.341 *	0.416 **	−0.331 *	−0.184	−0.142	0.376 *	0.051	0.367 *	0.096	0.093	0.028
Overall		0.295 **	0.229 *	−0.092	0.051	−0.063	0.058	0.157	0.164	−0.134	−0.051	0.004	0.076	−0.158	−0.027
With DD	EDT	0.347 **	0.229 *	−0.133	0.037	−0.132	0.063	0.219 *	0.204 *	−0.182	−0.108	−0.034	0.092	−0.112	0.084
Without DD		0.116	−0.245	−0.004	0.108	0.138	−0.049	0.001	0.011	0.022	0.124	0.027	0.050	−0.237	−0.284
Overall		0.020	0.091	0.081	0.127	0.149	−0.002	−0.024	−0.030	0.061	0.006	0.017	−0.085	0.041	−0.088
With DD	IVRT	0.130	0.082	0.089	0.145	0.150	−0.002	−0.034	−0.008	0.044	−0.036	0.051	−0.151	0.077	−0.039
Without DD		−0.233	0.026	0.009	0.081	0.192	−0.290	0.020	−0.195	0.104	0.161	−0.061	0.125	−0.054	−0.242
Overall		0.231 *	0.114	0.116	0.092	0.002	0.178	−0.192 *	0.114	0.193 *	−0.012	0.055	−0.045	−0.077	−0.115
With DD	LAVI	0.130	−0.019	0.157	0.159	0.056	0.063	−0.188	0.198	0.074	0.007	0.035	−0.018	−0.128	−0.206
Without DD		0.194	0.271	0.020	0.055	0.059	0.090	−0.169	−0.304	0.083	0.045	0.103	−0.113	0.070	0.028

* Statistically significant associations with *p* < 0.05. ** Statistically significant associations with *p* < 0.01. LVDD: left ventricular diastolic dysfunction; BMI: body mass index; WC: waist circumference; HbA_1c_: glycated hemoglobin; METS-IR: Metabolic Score for Insulin Resistance; AIP: atherogenic index of plasma; UACR: urinary albumin-to-creatinine ratio; TNF-α: tumour necrosis factor-alpha; hsCRP: high-sensitivity C-reactive protein; PLR: platelet-to-lymphocyte ratio; E: mitral E wave velocity (rapid filling) with pulsed Doppler; e’: mitral annular velocity with tissue Doppler imaging; EDT: E-wave deceleration time; IVRT: isovolumic relaxation time; LAVI: indexed left atrium volume.

**Table 6 jcm-14-05772-t006:** Associations between the echocardiographic parameters and other patient characteristics after 1-year follow-up, overall and in the groups with and without LVDD.

LVDD	Echocardiographic Parameters	Age	Diabetes Duration	Weight	CHA_2_DS_2_ -VASc score	HbA_1c_	Index C-Peptide	HDL- Cholesterol	CRI-I	CRI-II	AI	TNF-α	hs CRP
Overall		0.147	0.104	−0.048	0.300 **	0.083	0.049	0.072	0.032	0.061	0.032	0.294 **	−0.044
With DD	E/e’	−0.096	0.033	0.084	0.064	0.012	0.080	0.032	0.161	0.180	0.161	0.260 *	0.032
Without DD		0.266	−0.089	−0.222	0.525 **	0.184	−0.048	−0.030	−0.016	−0.035	−0.016	0.385 **	0.115
Overall		−0.429 **	−0.227 **	0.314 **	−0.381 **	−0.093	0.069	−0.152	0.192 *	0.192 *	0.192 *	−0.210 *	−0.040
With DD	E/A	−0.447 **	−0.301 *	0.262 *	−0.324 **	−0.205	0.087	−0.082	0.180	0.220 *	0.180	−0.273 *	0.033
Without DD		−0.260	0.000	0.343 *	−0.343 *	−0.205	0.084	−0.088	0.127	0.144	0.127	−0.023	−0.164
Overall		0.385 **	0.227 **	−0.050	0.096	−0.114	−0.008	0.131	−0.097	−0.137	−0.097	0.120	−0.115
With DD	EDT	0.416 **	0.290 *	−0.096	0.072	−0.007	0.012	0.064	−0.037	−0.073	−0.037	0.180	−0.236 *
Without DD		0.183	0.047	0.193	0.030	−0.277	0.125	0.125	−0.155	−0.237	−0.155	0.013	0.116
Overall		0.150	0.240 **	−0.032	−0.024	0.110	−0.179 *	0.074	−0.026	−0.024	−0.026	0.032	−0.193 *
With DD	IVRT	0.135	0.215	−0.046	−0.029	0.237 *	0.182	0.094	0.030	0.025	0.030	0.129	−0.097
Without DD		0.048	0.189	0.030	−0.204	−0.113	0.163	−0.046	−0.070	−0.072	−0.070	−0.138	−0.227
Overall		0.288 **	0.140	−0.006	0.121	−0.114	0.147	0.223 *	−0.054	−0.047	−0.054	0.162	−0.020
With DD	LAVI	0.189	−0.070	−0.095	0.054	0.134	0.200	0.223 *	−0.107	−0.120	−0.107	0.124	−0.001
Without DD		0.051	0.240	0.360 *	−0.313 *	−0.075	−0.015	−0.127	0.244	0.183	0.244	0.102	−0.054

* Statistically significant associations with *p* < 0.05. ** Statistically significant associations with *p* < 0.01; LVDD: left ventricular diastolic dysfunction HbA_1c_: glycated hemoglobin; CRI-I: Castelli risk index I; CRI-II: Castelli risk index II; AI: atherogenic index; TNF-α: tumour necrosis factor-alpha; hsCRP: high-sensitivity C-reactive protein; E: mitral E wave velocity (rapid filling) with pulsed Doppler; e’: mitral annular velocity with tissue Doppler imaging; EDT: E-wave deceleration time; IVRT: isovolumic relaxation time; LAVI: indexed left atrium volume.

## Data Availability

The data presented in this study are available on request from the corresponding author.

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
