# Peer review of "Age-Related Characteristics of Diastolic Dysfunction in Type 2 Diabetes Patients"

_jcm, 2025, doi:10.3390/jcm14165772_

Round 1
Reviewer 1 Report
Comments and Suggestions for Authors
Grigorescu et al. conducted a secondary analysis of cross-sectional and follow-up data to explore predictive factors for left ventricular diastolic dysfunction (LVDD) in a cohort of 138 outpatients with type 2 diabetes mellitus (T2DM). While the study addresses an important clinical issue, I have several concerns regarding the scientific rigor, clarity of methodology, and robustness of the findings.
- Inconsistency Between Abstract and Results
There appears to be a discrepancy in the descriptive statistics (e.g., participant age) reported in the abstract versus the main results section. This inconsistency should be addressed to ensure accuracy and clarity. The authors should clarify which figures are correct and revise accordingly to maintain internal consistency.
- Sample Size and Stratified Analyses
Given the modest sample size (n = 138), further stratification of results by age groups raises concerns about statistical power and the reliability of p-values. The rationale for stratifying such a limited sample should be justified more clearly, and the authors should acknowledge the increased risk of type I and type II errors due to small subgroup sizes.
- Multivariable Model Transparency
In the multivariable analysis for risk factor identification, it is unclear which covariates—beyond age—were included in the model. The current description of the modeling process is insufficient for replication. A clear explanation of variable selection criteria (e.g., based on univariate analysis, clinical relevance, or stepwise selection) and a list of included predictors should be provided to enhance methodological transparency.
- Statistical Methods for Repeated Measures
If both baseline and 1-year follow-up data were available, as implied, then the repeated-measures nature of the data should be considered in the analysis. For instance, some p-values reported in Table 2 appear to assume independent samples, which may not be appropriate. The authors should revise the analytical approach to account for within-subject correlations (e.g., using paired tests or mixed-effects models) and ensure that the interpretation of longitudinal changes is statistically valid.
- Limitations and Bias
The manuscript would benefit from a more thorough discussion of limitations. Specifically, potential sources of bias (e.g., selection bias, measurement error, confounding) should be acknowledged. Additionally, the authors should comment on the limited generalizability of the findings due to sample size, design constraints, and potential unmeasured confounders.
Author Response
Grigorescu et al. conducted a secondary analysis of cross-sectional and follow-up data to explore predictive factors for left ventricular diastolic dysfunction (LVDD) in a cohort of 138 outpatients with type 2 diabetes mellitus (T2DM). While the study addresses an important clinical issue, I have several concerns regarding the scientific rigor, clarity of methodology, and robustness of the findings.
We are grateful to the reviewer for the effort and the helpful suggestions. We applied all necessary modifications to the text, and we believe this has consistently improved the clarity and logic of the manuscript.
- Inconsistency Between Abstract and Results
There appears to be a discrepancy in the descriptive statistics (e.g., participant age) reported in the abstract versus the main results section. This inconsistency should be addressed to ensure accuracy and clarity. The authors should clarify which figures are correct and revise accordingly to maintain internal consistency.
Response 1: Thank you for your important comment. We corrected the error in the main result section (line 208).
- Sample Size and Stratified Analyses
Given the modest sample size (n = 138), further stratification of results by age groups raises concerns about statistical power and the reliability of p-values. The rationale for stratifying such a limited sample should be justified more clearly, and the authors should acknowledge the increased risk of type I and type II errors due to small subgroup sizes.
Response 2: Thank you for the recommendation. We included new information in section 2.4. (Statistical analyses) to confirm that non-parametric tests were used due to the sample size and data distribution. We also added a supplementary sentence related to statistical errors in the limitations section (lines 580-581).
- Multivariable Model Transparency
In the multivariable analysis for risk factor identification, it is unclear which covariates—beyond age—were included in the model. The current description of the modeling process is insufficient for replication. A clear explanation of variable selection criteria (e.g., based on univariate analysis, clinical relevance, or stepwise selection) and a list of included predictors should be provided to enhance methodological transparency.
Response 3: Thank you for this remarkable recommendation. We added the information about the logistic regression model (lines 180-189), as you suggested, and included a new paragraph within the results (lines 249-255).
- Statistical Methods for Repeated Measures
If both baseline and 1-year follow-up data were available, as implied, then the repeated-measures nature of the data should be considered in the analysis. For instance, some p-values reported in Table 2 appear to assume independent samples, which may not be appropriate. The authors should revise the analytical approach to account for within-subject correlations (e.g., using paired tests or mixed-effects models) and ensure that the interpretation of longitudinal changes is statistically valid.
Response 4: Thank you for this important recommendation. We included clarifications in the statistical analysis section, describing the tests we used for the repeated measures to compare the changes between baseline (T0) and follow-up (T1) for the whole sample and within each age group (lines 193-198).
- Limitations and Bias
The manuscript would benefit from a more thorough discussion of limitations. Specifically, potential sources of bias (e.g., selection bias, measurement error, confounding) should be acknowledged. Additionally, the authors should comment on the limited generalizability of the findings due to sample size, design constraints, and potential unmeasured confounders.
Response 5: Thank you for this important suggestion. A new paragraph (lines 591-596) has been introduced to discuss all these aspects.
Reviewer 2 Report
Comments and Suggestions for Authors
I reviewed with interest the manuscript by Elena-Daniela Grigorescu et al. "Age-Related Characteristics of Diastolic Dysfunction in Type 2 Diabetes Patients". In this article, the authors analyzed the presence of diastolic dysfunction in patients with diabetes mellitus over time - at baseline and one year later. As a result, the authors conclude that asymptomatic diastolic dysfunction of the left ventricle is common in patients with type 2 diabetes without atherosclerotic manifestations, and age is a significant risk factor for diastolic dysfunction. These observations may probably be of practical interest, but during the review I had the following questions and comments: 1. First of all, I have doubts about the novelty of the results obtained by the authors. Indeed, the authors' statements in the conclusion: "Asymptomatic left ventricular diastolic dysfunction is common in type 2 diabetes patients without atherosclerotic manifestations. Age is a significant risk factor for diastolic dysfunction" (lines 517-519) are well-known data, repeatedly published earlier. For which the authors conducted another study. 2. Interestingly, in some patients, the diastolic function of the left ventricle worsened after a year, while in others it increased. The authors do not analyze this fact in any way, although it is very interesting and in the future may help in finding ways to correct diastolic dysfunction in these patients. The authors only indicate that "No significant differences were found in the 52-week data between patients with worsened and those with stable or improved diastolic function" (lines 268-269); and repeat the same in the discussion ("In our study, a comparative analysis between patients with worsened and those with stable or improved diastolic function showed no significant differences after 52 weeks" - lines 487-489). I believe that the data from these groups should have been presented in full, including information on the therapy received. What is the reason for such different dynamics? This was more interesting for readers than information on the dynamics of the state of diastolic dysfunction of the LV in different age groups.
- In Table 12, some of the echocardiography indicators do not have units of measurement.
- Figure 1 is unclear, it needs to be corrected.
- The text contains a lot of digital information, which is difficult for the reader to digest. Thus, the data of the correlation analysis should be presented in a table.
- In addition, it is unclear why the authors decided to present such information as "The crosstabulation test revealed that more females had grade 1 LVDD than males (p = 0.04)" - lines 230-231. Why was the influence of gender assessed only for this indicator? It is unclear.
- The authors note: "The multivariate logistic regression analysis identified age over 65 as an independent risk factor for diastolic dysfunction (Exp B = 9.85, 95% CI = 1.29–75.36, p = 0.027)" - lines 232-233. In general, this is a completely expected fact, but the question is different - what indicators were included in the logistic regression model? Why is this analysis not mentioned in section 2.4. Statistical analyses?
Author Response
I reviewed with interest the manuscript by Elena-Daniela Grigorescu et al. "Age-Related Characteristics of Diastolic Dysfunction in Type 2 Diabetes Patients". In this article, the authors analyzed the presence of diastolic dysfunction in patients with diabetes mellitus over time - at baseline and one year later. As a result, the authors conclude that asymptomatic diastolic dysfunction of the left ventricle is common in patients with type 2 diabetes without atherosclerotic manifestations, and age is a significant risk factor for diastolic dysfunction. These observations may probably be of practical interest, but during the review I had the following questions and comments:
We thank the reviewer for the insightful comments on our manuscript. Based on the helpful suggestions we received, we made modifications that we trust will increase the global clarity and value of the manuscript.
- First of all, I have doubts about the novelty of the results obtained by the authors. Indeed, the authors' statements in the conclusion: "Asymptomatic left ventricular diastolic dysfunction is common in type 2 diabetes patients without atherosclerotic manifestations. Age is a significant risk factor for diastolic dysfunction" (lines 517-519) are well-known data, repeatedly published earlier. For which the authors conducted another study.
Response 1: Thank you for this remarkable comment. We have commented extensively on the necessity of a new/updated picture regarding diastolic dysfunction in elderly type 2 diabetes patients, given the current trend of increased prevalence of diabetes and cardiovascular complications, as well as the inconsistency in the literature (lines 597-608).
- Interestingly, in some patients, the diastolic function of the left ventricle worsened after a year, while in others it increased. The authors do not analyze this fact in any way, although it is very interesting and in the future may help in finding ways to correct diastolic dysfunction in these patients. The authors only indicate that "No significant differences were found in the 52-week data between patients with worsened and those with stable or improved diastolic function" (lines 268-269); and repeat the same in the discussion ("In our study, a comparative analysis between patients with worsened and those with stable or improved diastolic function showed no significant differences after 52 weeks" - lines 487-489). I believe that the data from these groups should have been presented in full, including information on the therapy received. What is the reason for such different dynamics? This was more interesting for readers than information on the dynamics of the state of diastolic dysfunction of the LV in different age groups.
Response 2: Thank you for this very constructive suggestion. We added a supplementary table (Table 4) to summarise the associations between categorical variables and the diastolic dysfunction status at the 1-year follow-up. We also included a new comment regarding the treatment effect on diastolic parameters (lines 549-551).
- In Table 12, some of the echocardiography indicators do not have units of measurement.
Response 3: Thank you for this observation. We added the units of measurement in Table 1.
- Figure 1 is unclear, it needs to be corrected.
Response 4: Thank you for the suggestion. We modified and clarified Figure 1.
- The text contains a lot of digital information, which is difficult for the reader to digest. Thus, the data of the correlation analysis should be presented in a table.
Response 5: Thank you for your recommendation. We added two supplementary tables (5 and 6), which summarise the results of the correlation analysis.
- In addition, it is unclear why the authors decided to present such information as "The crosstabulation test revealed that more females had grade 1 LVDD than males (p = 0.04)" - lines 230-231. Why was the influence of gender assessed only for this indicator? It is unclear.
Response 6: Thank you for this observation. We added the results regarding factors associated with grade 1 LVDD (lines 245-248).
- The authors note: "The multivariate logistic regression analysis identified age over 65 as an independent risk factor for diastolic dysfunction (Exp B = 9.85, 95% CI = 1.29–75.36, p = 0.027)" - lines 232-233. In general, this is a completely expected fact, but the question is different - what indicators were included in the logistic regression model? Why is this analysis not mentioned in section 2.4. Statistical analyses?
Response 7: Thank you for this very constructive observation. We added information about the logistic regression model in section 2.4 (lines 180-189) and included an explanatory paragraph at lines 249-255.
Round 2
Reviewer 1 Report
Comments and Suggestions for Authors
I’d personally suggest the authors double-check for consistency across their abstract, results, and discussion. Some interpretations don’t seem fully supported by the data.
It may also help if they clarify their rationale with more focus on age-related mechanisms from a hypothesis-driven perspective, rather than a purely data-driven one, which the current analysis doesn’t fully support.
Author Response
I’d personally suggest the authors double-check for consistency across their abstract, results, and discussion. Some interpretations don’t seem fully supported by the data.
We are grateful to the reviewer for the constructive feedback. We made extensive modifications to the text, and we signalled all additional sections in yellow (sections in green were modified during the previous revision). We also chose to delete some paragraphs, aiming to increase the consistency between various text fragments; we cross-marked those deletions and signalled them in blue. We hope all our changes helped the manuscript acquire better clarity and logic.
It may also help if they clarify their rationale with more focus on age-related mechanisms from a hypothesis-driven perspective, rather than a purely data-driven one, which the current analysis doesn’t fully support.
We revised the text in accordance with the reviewer's recommendations.
Reviewer 2 Report
Comments and Suggestions for Authors
The authors responded to my comments and made corrections to the text of the manuscript. I have no other comments.
Author Response
The authors responded to my comments and made corrections to the text of the manuscript. I have no other comments.
We thank the reviewer for the positive appreciation of our manuscript. Based on the suggestions from the other reviewer, we made some changes that we trust will increase the global clarity and value of the manuscript.